



# Greenhouse gases modulate the strength of millennial-scale subtropical rainfall, consistent with future predictions

Fei Guo[1,2,3*], Steven C. Clemens[2,*], Yuming Liu[1,4], Ting Wang[1,4], Huimin Fan[1], Xingxing Liu[1],

Youbin Sun[1,5,6]

[1]State Key Laboratory of Loess and Quaternary Geology, Institute of Earth Environment, Chinese

Academy of Sciences, Xian 710061, China.

[2]Department of Earth, Environmental, and Planetary Sciences, Brown University, Providence, RI

02912-1846, USA

[3]Institute of Marine Science and Technology, Shandong University, Qingdao 266237, China

[4]University of Chinese Academy of Sciences, Beijing 100049, China

[5]CAS Center for Excellence in Quaternary Science and Global Change, Xian 710061, China.

[6]Open Studio for Oceanic-Continental Climate and Environment Changes, Pilot National

Laboratory for Marine Science and Technology (Qingdao), Qingdao 266200, China.

Corresponding author: Fei Guo (guofei@ieecas.cn) and Steven C. Clemens

(steven_clemens@brown.edu)

**Highlights**

The new precipitation-sensitive proxy (Ca/Ti) shows persistent millennial-scale East Asian summer monsoon changes over past 650 ka;

The magnitude of millennial-scale variability is modulated by AMOC at the eccentricity and precession bands.

Increasing GHG and strong insolation lead to more frequent occurrence of extreme rainfall, consistent with model results.



**Abstract:** Millennial-scale East Asian monsoon variability is closely associated with natural hazards through long-term variability in flood and drought cycles. Therefore, exploring what drives the millennial-scale variability is of significant importance for future prediction of extreme climates. Here we present a new East Asian summer monsoon (EASM) rainfall reconstruction from the northwest Chinese loess plateau spanning the past 650 ka. The magnitude of millennial-scale variability (MMV) in EASM rainfall is linked to ice volume and greenhouse gas (GHG) at the 100-kyr earth-orbital eccentricity band and to GHG and summer insolation at the precession band. At the glacial-interglacial cycle, gradual changes in $CO_2$ at times of intermediate ice volume leads to increased variability in North Atlantic stratification and Atlantic meridional overturning circulation, propagating abrupt climate changes into East Asia via the westerlies. Within the 100-kyr cycle precession variability further enhances the response, showing that stronger insolation and increased atmospheric GHG cause increases in the MMV of EASM rainfall. These findings indicate increased extreme precipitation events under future warming scenarios, consistent with model results.

**Key words:** EASM rainfall, MMV, GHG modulation, precession band

## 1. Introduction

Chinese loess is a unique terrestrial archive that can well document East Asian monsoon (EAM) variability at tectonic to millennial timescales (Porter and An, 1995; An et al., 2011). High-resolution loess records have revealed persistent millennial-scale (1-10 kyr periodicity) EAM fluctuations spanning the last several glacial cycles (Guo et al., 1996; Sun et al., 2012, 2021a,b; Guo et al., 2021), which are dynamically linked with high-latitude abrupt changes in the



north Atlantic including Heinrich (H) (Heinrich, 1988) and Dansgaard-Oeschger (DO) events
(Dansgaard et al., 1982). This millennial-scale monsoon variability is superimposed on
glacial-interglacial variations (Ding et al., 1999; Clemens et al., 2018). Abrupt summer monsoon
changes are closely linked to natural hazards such as flood and drought events (Huang et al., 2007),
since the summer monsoon plays a leading role in transporting water vapor from low to
middle/high latitudes of the northern hemisphere (Liu et al., 2013; An et al., 2015). Abrupt rainfall
events associated with short-term summer monsoon variations strongly influence agriculture, food
production, water supply and social economic development (Huang et al., 2007; Cook et al., 2010;
Li et al., 2017). However, how these flood/drought events are affected by both natural and
anthropogenic factors remains poorly constrained. Understanding the mechanisms that modulate
the magnitude of millennial-scale variability (MMV) is of critical importance for the scientific
community as well as policy makers. Here we use the term "modulate" in the context of
lower-frequency components of the climate system influencing or determining the amplitude of a
higher-frequency components.
A number of well-dated, high-resolution speleothem $\delta^{18}O$ records have been developed in
recent years (Wang et al., 2008; Cheng et al., 2016), providing the opportunity to examine the
underlying relationship(s) between East Asian monsoon MMV and potential longer-term
(orbital-scale) modulators. The latest research suggests that the MMV through the Pleistocene is
influenced by both glacial boundary condition and orbital configurations (Sun et al., 2021b).
Cheng et al., (2016) hypothesized, on the basis of an East Asian composite speleothem $\delta^{18}O$
record ($\delta^{18}Osp$), that periods of maximum Northern Hemisphere summer insolation correspond to
weaker millennial-scale variability. Subsequently, however, Thirumalai et al (2020) showed that



precession does not modulate the MMV of $\delta^{18}Osp$ and postulated that it is, instead, modulated by
internal processes related to the cryosphere. This work also raised the possibility that $\delta^{18}Osp$ is
decoupled from regional Asian monsoon rainfall over millennial timescales (Zhang et al., 2018).
As such, two important outstanding questions remain; is there a reliable proxy for East Asian
summer monsoon (EASM) rainfall at the millennial timescale and what factors modulate the
MMV thereof?

Due to weak pedogenesis and high sedimentation rates, millennial-scale oscillations are well

preserved in the western and northwestern Chinese Loess Plateau (CLP) over the past glacial
cycles (Sun et al., 2012, 2021a; Guo et al., 2021). The Linxia profile is well-suited for
reconstructing rapid monsoon changes because it is located in monsoon frontal zone and sensitive
to high- and low-latitude climate variability. To address the above questions, we have generated a
high-resolution summer monsoon proxy (Ca/Ti) from Linxia on the western CLP (Fig. 1). The
Ca/Ti ratio is a precipitation-sensitive proxy linked to summer monsoon rainfall (Guo et al., 2021).
Low values of Ca/Ti indicate stronger Ca leaching associated with intensified summer rainfall.
The new precipitation proxy (Ca/Ti) and $\delta^{18}Osp$ are evaluated to elucidate the modulating drivers
of these two proxy records. As discussed in the Results section, we find that the MMV of Ca/Ti is
mainly modulated by ice volume and greenhouse gases (GHG) at the eccentricity band. Both
GHG and summer insolation modulate the MMV of Ca/Ti at the precession band but not that of
$\delta^{18}Osp$; $\delta^{18}Osp$ MMV is modulated by winter insolation at the eccentricity and obliquity bands.
The interpretations of these results are presented in the Discussion section.
**2. Materials and Methods**

The Linxia (LX; 103.63°E, 35.15°N, 2,200 m a.s.l.) loess record is from the western edge of the



CLP (Fig. 1). At present, mean annual temperature and precipitation in this region are about 8.1°C
and 484 mm, respectively, with ~80% of the annual precipitation falling during the summer season
(May to September). The 203.8 m-long core A (LXA) consists of 185 m of eolian loess-paleosol
sequences, underlain by 17 m of fluvial loess and 1.8 m of sandy gravel layers. The 72 m-long
core B (LXB) and a 7 m pit were excavated in 2017. Powder samples were collected at 2 cm
intervals for analyzing mean grain size (MGS). As well, 2-cm resolution samples were dried at
40°C overnight and ground to 200 mesh size (about <75 μm) with an agate mortar and pestle, and
then pressed into a plastic sheet (4 cm × 4 cm × 0.3 cm), creating a flat and homogeneous slide.
The plastic slides were then placed on a wood pallet for XRF scanning to obtain elemental
intensities (Guo et al., 2021).
The upper 18 m is mapped to the OSL-dated Yuanbao loess outcrop (~4 kilometers away) (Lai
et al., 2006). The chronostratigraphy has been generated using an independent loess chronology
generated by synchronizing Chinese loess and speleothem $\delta^{18}O$ records back to 650 ka (Sun et al.,
2021a). The first set of control points tie the loess/paleosol boundaries $S_6$ to $S_0$ to the timing of the
glacial terminations/inceptions in speleothem $\delta^{18}O$ (Cheng et al., 2016). The second and third sets
of age control points tie the timing of precessional transition boundaries and abrupt cooling events
in the MGS record to those in speleothem $\delta^{18}O$ (Fig. 2), based on the assumption that the East
Asian summer and winter monsoon co-vary with each other at orbital timescales, and
millennial-scale abrupt events are synchronous in the northern hemisphere (Hemming et al., 2004;
Sun et al., 2012; Clemens et al., 2018). The tie points are shown in Fig. 2.
In order to estimate the MMV, all the raw datasets are linear interpolated at 0.1 kyr interval. The
original time series are filtered using a Butterworth filter at a cutoff threshold of 10 kyr (e.g.



Ca/Ti-hi-10kyr). The moving standard deviation of millennial-scale variability is calculated to
ascertain the orbitally-related modulation and its association with internal and external forcing
using 2 kyr sliding window (calculation method follows Thirumalai et al., 2020). The spectra of
all       proxies       were       calculated       using       the       Lomb-Scargle       periodogram
(https://exoplanetarchive.ipac.caltech.edu/cgi-bin/Pgram/nph-pgram), which has the advantage of
analyzing discontinuous time series and removal of spurious spectral characteristics (VanderPlas,
2018). Normalized and combined orbital parameters eccentricity, tilt, and negative precession
(ETP), GHG, insolation, and benthic $\delta^{18}O$ were evaluated by wavelet coherence (WTC) to extract
maximal phase and amplitude correlations with astronomical, ice volume and greenhouse gases
forcing over the past 650 ka. WTC between time series was performed in a Monte Carlo
framework (n = 1, 000) following Grinsted et al., (2004). The WTC would help to detect the
period in different frequency bands where the two time series co-vary (but does not necessarily
have high power). The black arrows in the figures represent the phrase relationship between the
two time sequences with rightward, upward and downward arrows indicate the in-phrase, leading
and lagging phrase, respectively. The colour scale indicates the amplitude correlations between the
two datasets.

In this paper, the parameter $\Delta RF_{GHG}$ is regarded as GHG radiative forcing (GHG RF, using the

GHG RF instead of $\Delta RF_{GHG}$ in the discussion section) and applied in WTC to evaluate the
relationship between MMV of Ca/Ti and $\delta^{18}O$ sp. The $\Delta RF_{GHG}$ is reconstructed by referencing the
content of EPICA ice core greenhouse gases to the modern value.
$\Delta RF_{GHG}$ is defined as the difference between a certain past GHG level ([$CO_2$] and [$CH_4$]) and the
pre-industrial greenhouse gas level ([$CO_2$]$_0$ = 280 ppm, [$CH_4$]$_0$ = 700 ppb) (Ramaswamy et al.,

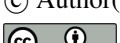



2001). While $CH_4$ contributes <5%, we calculated the $\Delta RF_{GHG}$ using both $CO_2$ and $CH_4$. The
equation used to determine $\Delta RF_{GHG}$ is as follows (Lo et al., 2017):
$\Delta RF_{GHG} = \Delta RF_{CO2} + \Delta RF_{CH4}$
$= 4.841\ln([CO_2]/[CO_2]_0) + 0.0906(\sqrt{[CO_2]} - \sqrt{[CO_2]_0}) + 0.036\ln(\sqrt{[CH_4]} - \sqrt{[CH_4]_0})$

**3. Results**

The MGS reflects grain-size sorting and is very sensitive to winter monsoon variations (Porter
and An, 1995; Sun et al., 2006) with larger particle size during the glacials. The Ca/Ti ratio
reflects precipitation-induced leaching intensity linked to summer monsoon rainfall (Guo et al.,
2021), with lower value during the interglacials. The high resolution $\delta^{18}O$ of Sanbao-Hulu
speleothem is an indicator of EASM changes at orbital to centennial timescales (Cheng et al.,
2016). The MGS and Ca/Ti exhibit distinct glacial-interglacial and precessional variations over the
last 650 ka as seen in LR04 $\delta^{18}O$ (Lisiecki and Raymo, 2005) and speleothem $\delta^{18}O$ (Cheng et al.,
2016), respectively (Fig. 2).
Both Ca/Ti and $\delta^{18}O_{sp}$ records show clear millennial-scale fluctuations overlaying orbital-scale
variations. The high frequency millennial signals (Materials and Methods) persist over the last 650
ka, but the amplitude varies from proxy to proxy (Fig. 3a and S1a). Spectral analysis of the raw
records and MMV for loess and speleothem records display variable associations with
eccentricity- (~100 kyr), obliquity- (~41 kyr), and precession-scale (~23 and ~19 kyr) over the
past 650 ka. Loess Ca/Ti variance is mainly concentrated in obliquity with lesser variance in the
eccentricity and precession bands (Fig. 3b), indicating prominent ice volume (eccentricity and
obliquity) and isolation (precession) forcing. The speleothem $\delta^{18}O$ shows predominant
precession-scale variance (Fig. S1b) suggesting strong links to insolation forcing (Cheng et al.,





2016). These results indicate ice volume and insolation play dominant roles in driving changes in
loess Ca/Ti and speleothem $\delta^{18}$O, respectively (Cheng et al., 2016; Sun et al., 2021a).
Millennial-scale fluctuations co-exist with long-term orbital- and ice-volume variability; we
seek to assess the potential linkages among them and in particular, the extent to which MMV is
modulated by these longer-term orbital and internal climate parameters. The spectra of Ca/Ti
MMV shows dominant eccentricity with less strong precession and weak obliquity variance (Fig.
3d). The spectrum of $\delta^{18}$Osp MMV has a small peak near 100 kyr and an offset 41 kyr peak with
little to no variance at the 23 kyr period (Fig. S1d). Thus, while both proxies are similarly
modulated at the 100-kyr period (such that the MMV is larger during glacial intervals relative to
interglacial times) the MMV modulation is variable for the two proxies at other orbital bands.
As with the spectral differences in the raw records, the MMV spectra also implies different
MMV modulating drivers, potentially associations with insolation, ice volume, and/or GHG
(Thirumalai et al., 2020). How do internal and external drivers interact with each other and
modulate the MMV of these records at the orbital timescale? We performed wavelet coherence
and phase analyses of both MMV records relative to ETP, ice volume, $\Delta RF_{GHG}$, summer insolation,
and winter insolation to identify which variables might modulate the MMV of these EASM
records. The MMV in Ca/Ti is strongly coherent with ice volume and GHG at the 100,000-year
earth-orbital eccentricity band and with GHG and summer insolation at the 23,000-year precession
band (Figure 4c, d, g). $\delta^{18}$Osp MMV is most strongly coherent with GHG and ice volume at the
100-kyr band and with winter insolation at the eccentricity and obliquity bands (Figure S2c, d, g).
**4. Discussion**
**4.1 Orbital-scale modulation factors for MMV of the EASM**



Previous geological records and modeling indicate that high latitude ice volume or ice sheet
topography play important roles in triggering abrupt climate changes (Broecker et al.,1994; Clark
et al., 2001). In particular, abrupt climate changes are highly sensitive to ice volume variations; ice
sheets are widely hypothesized to motivate and amplify these high frequency signals within a
constrained benthic oxygen isotope-"ice volume threshold" between 3.5 and 4.5‰ (Bailey et al.,
2010; Naffs et al., 2013; Zhang et al., 2014). Wavelet coherence between the MMV of loess Ca/Ti,
speleothem $\delta^{18}O$ and the global benthic $\delta^{18}O$ stack show excellent coherence and near-zero phase
with ice volume at the 100 kyr band (Fig. 4e, g and S2e, g); this in-phase variation demonstrates
that EASM MMV primarily follows the glacial-interglacial rhythm of ice volume variations,
enlarged during glacial times and dampened during interglacial times. However, coherence of the
MMV for these two proxies with the benthic $\delta^{18}O$ stack is relatively weak and variable at the 41
kyr band ($\delta^{18}Osp$; Figure S2e, g) and 23-kyr band (Ca/Ti; Fig. 4e, g). These relationships
demonstrate that ice volume directly modulates the MMV of the EASM, predominantly at the 100
kyr band, with high ice volume corresponding to larger MMV.
GHG concentration is another potential driver of abrupt climate changes (Alvarez-Solas et al.,
2011; Zhang et al., 2017).Wavelet coherence between the MMV of loess Ca/Ti, speleothem $\delta^{18}O$
and the record of GHG RF show excellent coherence and ~180° phase at the 100-kyr eccentricity
band (Fig. 4b, d and Fig. S2b, d) indicating strong MMV at times of low GHG. Given the coupled
nature of global ice-volume and atmospheric GHG, it is clear that over the late Pleistocene
glacial-interglacial cycles, these two factors modulate the MMV of the EASM as recorded by
Ca/Ti and speleothem $\delta^{18}O$ such that abrupt climate change is amplified during times of high ice
volume and low GHG concentration. However, this is not the case for the precession band. MMV



of loess Ca/Ti displays discrete intervals high coherence and near-zero phase with GHG RF at the
precession band (Figure 4b, d), which is not the case for speleothem δ¹⁸O (Figure S2b, d). Thus,
GHG RF does play a role in modulating Ca/Ti MMV but not that of δ¹⁸Osp at the precession band,
indicating a difference in the millennial-scale response of these two proxies at this time-scale. We
investigate this further by assessing the response to local insolation forcing.
The MMV of Ca/Ti show discontinuous relatively weak coherence with 35°N summer
insolation at the precession band with even weaker coherence at the 41-kyr band (Figure 4a, c);
we note that the summer insolation modulation is less strong relative to that of GHG at the
precession band (Figure 4b, d). In contrast, the MMV of δ¹⁸Osp displays high coherence and zero
phase with 35°N winter insolation at 100 kyr period, relatively weaker coherence, with a lagging
phase, at the 41 kyr band, and negligible coherence at the 23-ky band (Figure S2a, c). These
results indicate that the MMV of speleothem δ¹⁸O is modulated by local winter insolation,
opposite to the Cheng et al., (2016) hypothesis calling on north hemisphere summer insolation.
**4.2 Mechanism and implication for modulation of EASM MMV**
At the glacial-interglacial timescale, the MMV is amplified under the glacial boundary
conditions. This indicates dynamic linkages with high latitude North Atlantic Heinrich and DO
events (Cheng et al., 2016; Sun et al., 2012, 2021a, b). Heinrich and DO variability are linked to
Northern Hemisphere ice sheet (NHIS) perturbations via its influence on fresh-water flux into the
North Atlantic Ocean and consequent Atlantic meridional overturning circulation (AMOC)
changes (McManus et al., 1999; Hemming, 2004; Naffs et al., 2013). At times of intermediate ice
sheet volume, minor changes in NHIS height and atmospheric $CO_2$ concentrations can trigger the
rapid climate transitions (Zhang et al., 2014, 2017). Altering the height of NHIS leads to changes



in the gyre circulation and sea-ice coverage by shifting the northern westerlies (Zhang et al., 2014).
The maximum westerly wind stress shifts northwards associated with gradual increase of the
Northern Hemisphere ice volume. The northward westerly, in turn, encourages the EASM rain
belt to move northward (He et al., 2021) and results in increases in the MMV of EASM rainfall
(especially northern China). In addition, $CO_2$ acts as an internal feedback agent to AMOC changes
(Barker et al., 2016). Under intermediate glacial condition, when the AMOC reaches a regime of
bi-stability, rising $CO_2$ during Heinrich Stadial cold events can trigger abrupt transitions to warm
conditions. Decreasing $CO_2$ during warm events leads to abrupt cooling transitions (Zhang et al.,
2017). Therefore, $CO_2$ generally provides a negative feedback on MMV of EASM rainfall. During
interglacial times decreasing ice volume, accompanied by reduced sea ice and more frequent
freshwater perturbation, is correlated with lower frequency and smaller amplitude variability in
abrupt climate events. The co-evolving GHG concentrations would further alter the sea surface
temperature by greenhouse forcing, subsequently modulating the MMV.
Within the 100,000-year cycle, precession-band variability (4-5 cycles), characterized by
increased insolation and atmospheric GHG, further heightens the positive response, leading to
larger MMV of subtropical rainfall. Recent transient sensitivity experiments suggests that
millennial-scale rainfall variability is driven primarily by meltwater and secondarily by insolation
(He et al., 2021). During interglacial times under the combined influence of insolation and $CO_2$,
model simulation shows that when insolation reaches the lower "threshold" value (between 358.2
and 352.1 W. m$^{-2}$), it triggers a strong abrupt weakening of the AMOC and results in abrupt
cooling transitions over last 800 ka (Yin et al., 2021). Increased insolation could warm sea surface
temperature and accelerate freshwater input from high latitude ice sheet as well as altering GHG



concentration in the atmosphere (Lewkowicz and Way, 2019), which could, in turn, modulate
MMV changes in the low latitude monsoon regions.

If both millennial-scale Ca/Ti and $\delta^{18}O_{sp}$ represent subtropical rainfall amount, the

modulation factors should be consistent. However, eccentricity, obliquity and precession bands
MMV modulators differ for loess Ca/Ti and $\delta^{18}O_{sp}$, indicating they monitor different aspects of
millennial-scale monsoon circulations. Modern observations and Lagrangian trajectories of air
parcels in China during the summer monsoon indicate that moisture-induced precipitation doesn't
derive from the strongest water vapor pathways (Sun et al., 2011; Jiang et al., 2017); local water
vapor recycling contributes significantly to regional precipitation in East China (over 30%) and
North China (exceeding 55%) (Shi et al., 2020). Hence, we speculate that $\delta^{18}O_{sp}$ MMV monitors
changes in the isotopic composition of rainfall, varying with changes in westerly transport paths
associated with North Atlantic cooling events, consistent with the MMV of $\delta^{18}O_{sp}$ being closely
linked to winter insolation at 100- and 41- kyr periods and the absence of MMV modulation at
precession band. We further hypothesize that Ca/Ti mainly represents the MMV in local rainfall
amount, consistent with the MMV of tropical rainfall being more dynamically related to GHG and
summer insolation at precession band.

In recent decades atmospheric GHG concentration is accelerating due to anthropogenic

contribution of fossil fuels, suggesting that EASM (extreme) precipitation will increase as well.
This inference is consistent with model simulations indicating that the number of extreme daily
precipitation events and mean precipitation overall will increase significantly in response to higher
GHG concentration (Dairaku and Emori, 2006). The anthropogenic GHG-evoked warming is
projected to increase the lower-tropospheric water vapor content and enhance the thermal contrast





between land and ocean (Kitoh et al., 1997). This will give rise to a northward shift of lower
tropospheric monsoon circulation and an increase rainfall during the East Asian summer monsoon
(Vecchi and Soden, 2007). Our results indicate that factors modulating EASM precipitation MMV
in the past are consistent with those predicted to influence future changes in monsoonal
precipitation, lending further confidence in those projections.
**5. Conclusions**
Our high-resolution loess Ca/Ti record displays millennial monsoon oscillations that persist
over the last 650 ka. Wavelet results highlight remarkable GHG modulation at both 100 kyr and
precession band as well as ice volume at 100 kyr period and local insolation forcing at precession
band. The MMV of loess Ca/Ti and speleothem $\delta^{18}O$ are modulated by different orbital factors,
implying that these two proxies document different climatic response of millennial-scale monsoon
circulation. The inferred mechanism of how these internal and external factors modulate the MMV
calls on dynamic linkages to variability in AMOC at both eccentricity and precession bands. In
recent decades, atmospheric GHG concentration is dramatically increasing due to anthropogenic
contribution of fossil fuels (Bousquet et al., 2006; Davis et al., 2010), resulting in accelerated
melting of ice-sheets in bi-polar regions (Swingedouw et al., 2008; Golledge et al., 2019). Their
combined effects lead to more frequent occurrences of extreme rainfall (Dairaku and Emori, 2006;
IPCC, 2018). Our results indicate that the MMV EASM rainfall is modulated by ice volume, GHG,
and insolation factors, consistent with those predicted to influence future changes in monsoonal
precipitation.

**CRediT authorship contribution statement**



Fei Guo mainly contributes to the experiments, data analysis, idea and draft paper. Prof.
Steven Clemens and Youbin Sun help to design the program and revise the draft. Huimin Fan
assists to perform the experiments and data processing. Ting Wang, Yuming Liu and Xingxing Liu
make contributions to the fieldwork and paper discussion.

**Acknowledgments**
We thank Xiaojing Du for offering idea on potential model test for this paper. This work was
supported by grants from the Strategic Leading Research Program of Chinese Academy of Science
(XDB40000000) and National Natural Science Foundation of China (41525008 and 41977173).

**Declaration of Competing Interest**
The authors declare that they have no known competing financial interests or personal
relationships that could have appeared to influence the work reported in this paper.

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

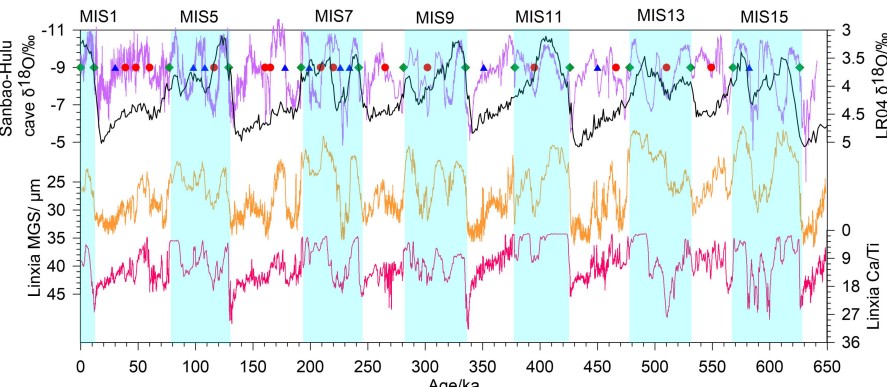

461

**Figure 2** Variations of mean grain size, Ca/Ti over last 650 ka and age model of Linxia loess

section. Comparison of mean grain size and Ca/Ti in Linxia section with Sanbao-Hulu (Cheng et

al., 2009, 2016) and benthic δ¹⁸O stack (Lisiecki and Raymo, 2005). The dark brown squares, blue

triangles and red dots represent the first (glacial-interglacial transition), second (precession cycles)

and third (millennial-scale events) class age control points at the corresponding position of cave

record, respectively (Sun et al., 2021a). Light blue bands donate the interglacial times.




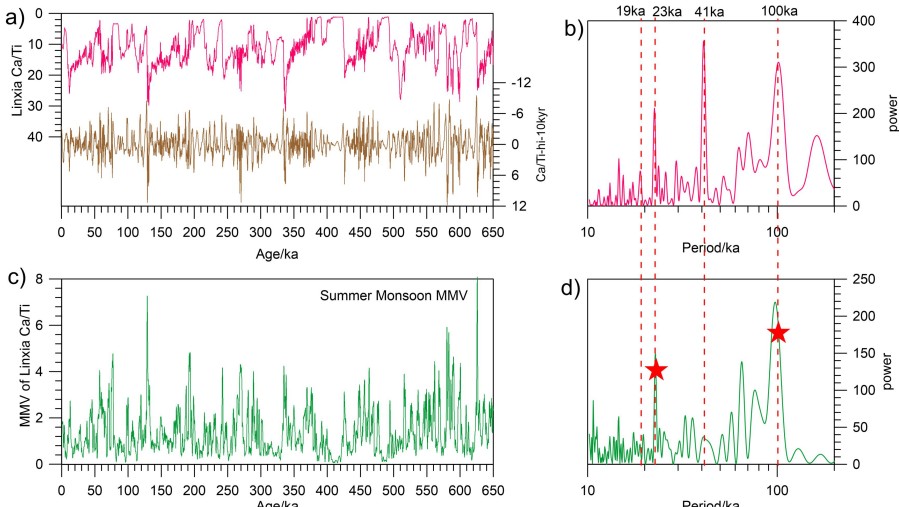


**Figure. 3** Raw datasets, millennial-scale components (10 kyr high pass filtering signals) and
MMV of the Linxia loess Ca/Ti record over the past 650 ka with their corresponding spectra. The
orbital bands are marked with red dashed lines (eccentricity-100 kyr, obliquity-41
kyr ,precession-23 kyr and 19 kyr). Clearly variable eccentricity, obliquity and precession
variances as well as persistent millennial-scale components are observed for loess Ca/Ti and
MMV.



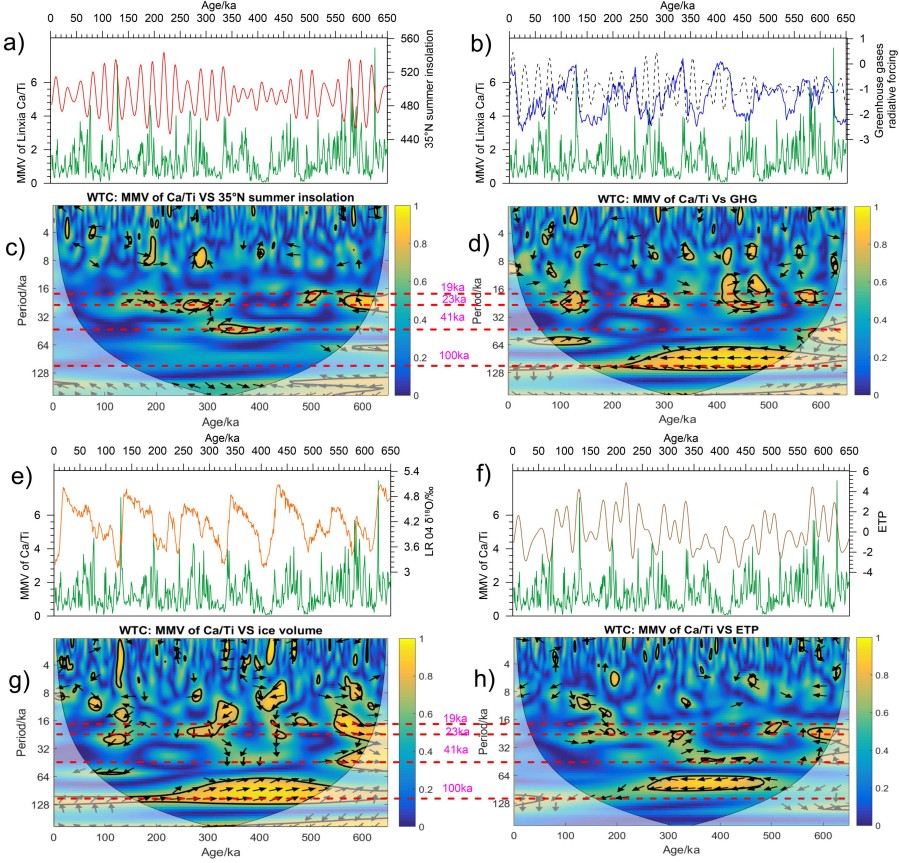

**Figure. 4** Comparison of a) 35°N summer insolation forcing, b) GHG radiative forcing (black dish

line donates the precession band-pass filtering results of $\Delta RF_{GHG}$) and e) ice volume and f) ETP

for MMV of Linxia loess Ca/Ti; Wavelet coherence between c) 35°N summer insolation, d) GHG

radiative forcing, g) ice volume, h) ETP and MMV of loess Ca/Ti over the past 650 ka. The orbital

bands are marked with red dashed lines (eccentricity-100 kyr, obliquity-41 kyr , precession-23 kyr

and 19 kyr). The orange color indicates strong correlation for the two time series. The black lines

plot coefficients of determination is more than 0.76. The black arrows represent the phrase

relationship with rightward, upward and downward arrows indicating in-phrase, leading and

lagging phrase, respectively. Strong eccentricity, weak obliquity and precession bands ice volume

modulation are observed for MMV of loess Ca/Ti. Strong eccentricity and precession bands GHG

modulation as well as weak summer insolation forcing are detected for MMV of loess Ca/Ti.