# Peer review of "Greenhouse gases modulate the strength of millennial-scale"

_Climate of the Past, 2021_

## Author Comment (AC1)

Review Guo et al.,

Guo et al., generated a high-resolution summer monsoon proxy (Ca/Ti) from Linxia on the western CLP. The Ca/Ti ratio is interpreted as a precipitation-sensitive proxy linked to summer monsoon rainfall (Guo et al., 2021). The new precipitation proxy (Ca/Ti) and an East Asian composite speleothem $\delta^{18}O$ record ($\delta^{18}Osp$) are evaluated to elucidate the modulating drivers of these two proxy records.

The authors find that the MMV of Ca/Ti is mainly modulated by ice volume and greenhouse gases (GHG) at the eccentricity band. Both GHG and summer insolation modulate the MMV of Ca/Ti at the precession band but not that of $\delta^{18}Osp$; $\delta^{18}Osp$ MMV is modulated by winter insolation at the eccentricity and obliquity bands. The inferred mechanism of how these internal and external factors modulate the MMV calls on dynamic linkages to variability in AMOC at both eccentricity and precession bands.

Results suggest that the MMV EASM rainfall is modulated by ice volume, GHG, and insolation factors, consistent with those predicted to influence future changes in monsoonal precipitation.

This is an interesting study in which the authors address two important outstanding questions: 1) is there a reliable proxy for East Asian summer monsoon (EASM) rainfall at the millennial timescale and 2) what factors modulate the MMV thereof?

The paper is clear and well wrote, and is suitable for a journal such as Climate of the Past. However the authors must answer to major/minor comments (see below) to be sure that the main conclusions of their paper can be fully supported before considering publication.

Re: Thanks for reviewing our paper and giving us useful advice to improve our manuscript.

Major comment:

- There is no Figure with the age control points that include the error bars on these control points and more generally no errors for the age model used in this study. I suggest to add a figure that include the depth/age and errors for the age model of the loess record.

Re: Thanks for reminding us of adding a figure to show the error bars of our age model. The detail information of the age model was published in another journal (Sun et al., 2021, Earth-Science Reviews). We would mention this messages in method section. The following Figure 1 shows errors of our age model and help to explain the age model error would not affect the results of the wavelet coherence analysis. We use two different targets to establish our age model. These grain-size time series yield good correlation between loess/paleosol boundaries and glacial/interglacial transitions. The age differences between the two age models for most glacial terminations and precession cycles are around 2-4 kyr (Sun et al., 2021, Figure 1).

Sun, Y., Clemens, S., Guo, F., Liu, X., Wang, Y., Yan, Y., and Liang, L.: High-sedimentation-rate loess records: A new window into understanding orbital-and millennial-scale monsoon variability, Earth-Sci. Rev., 103731, https://doi.org/10.1016/j.earscirev.2021.103731, 2021a.

[Figure]

Figure 1 a) Down-core variations of mean grain size and sedimentation rate against depth, dark blue dots represent the age control points at the corresponding depth of mean grain size. b) Comparison of mean grain size in Linxia section with Sanbao-Hulu (Cheng et al., 2009, 2016) and benthic δ18O records (Lisiecki and Raymo, 2005), colorful dots represent age control points at the corresponding depth of cave record. Light blue bands donate the interglacial times. c) The comparison between the grain size based age model (Purple line, control points in red) and loess-cave comparison based age model (orange line, control points in black) over last 650 ka. Light blue and gray bars indicate age difference between two comparable time series.

What is the implication of the age model errors for the wavelet coherence correlations that authors conducted (against GHG, ETP, Insolation and benthic δ18O on Figure 4) and for the millennial-scale component extraction (Figure 3) ?

Re: That's really a good question. We should take the influence of age model error into consideration in method section. We assessed the this kind of influence and add the result in method parts. We isolated the millennial-scale components of loess Ca/Ti at the different age model for comparison (Figure 2). The age model difference of millennial-scale variability is 1-3

kyr. We applied Ca/Ti on grain size based age model to calculate the MMV and conduct wavelet coherence analysis, showing almost the same correlation and phase at the orbital frequency bands (Figure 3) as that of WTC results at loess-cave comparison based age model (manuscript Figure 4).

[Figure]

Figure 2 The comparison of millennial-scale loess Ca/Ti variations on grain size based age model (purple line) and loess-cave comparison based age model (orange line).

[Figure]

Fig. 3 Comparison of a) ice volume, b) GHG forcing, c) ETP and d) local summer insolation with MMV of loess Ca/Ti over the past 650 ka at the grain size based age model.

Minor comments:

- What is the resolution of the Ca/Ti record (in years) before resampling? I could not find it in the text.

Re: Sorry for forgetting to add the resolution of our new proxy. We revised it accordingly. The powder samples were scanned at 2 cm interval, with the resolution ranging 50~200 yr (Figure 1).

- Introduction part (lines 51-60) : "flood and drought events". What is the definition of the authors here for flood and drought events? And at which time scale this events occur? Are they directly related to the millennial scale variability the authors reconstruct in this paper?

Re: In this paper, the definition of flood and drought events are centennial to decade (100-10 yr) timescale natural disaster.

---

## Author Comment (AC2)

This paper introduce a new East Asian summer monsoon rainfall reconstruction from the northwest Chinese loess plateau over the last 650,000 years. The new precipitation proxy (Ca/Ti) and speleothem $\delta^{18}O$ ($\delta^{18}Osp$) are assessed to illustrate the modulating drivers of magnitude of millennial monsoon variability (MMV) at the orbital timescale. Wavelet analysis highlights the remarkable ice volume and GHG modulation at 100 kyr band as well as GHG and local insolation forcing at precession band for the MMV of Ca/Ti, but not that of MMV in $\delta^{18}Osp$. The MMV of loess Ca/Ti and $\delta^{18}Osp$ are modulated differently at orbital time scales, implying that these two proxies document different climatic response of millennial-scale monsoon circulations. At the precession band, increasing atmospheric GHG following with larger insolation results in further enhancement in MMV of EASM rainfall, which agrees with the model result and prediction in more frequently occurrence of extreme rainfall under future global warming conditions.

In general, the content of their paper is very interesting. They link the interactions between millennial-scale variability and orbital-scale driving factors. Based on their findings, they predict how would millennial-scale abrupt East Asian summer monsoon rainfall would evolve under future warming conditions. Some suggestion are as follows:

One Highlight of this paper is the new summer rainfall indicator Ca/Ti. However, they do not mention too much about how Ca/Ti link to the East Asian summer rainfall and associated mechanism.

Re: Thanks for reminding us of highlighting our new proxy. We would add relative contents in last paragraph of introduction section. See new revised version.

In the introduction section, the logic of each paragraph could make some changes for improvements.

Re: We would like to check logic of each paragraph and make some changes for the introduction part.

In results section, the content includes some discussion sentences.

Re: Yes, the last paragraph of result section includes some discussion. These sentences are applied to introduce scientific questions that we want to discuss in next section.

---

## Author Comment (AC3)

The paper by Guo et al. submitted to CP is based on a new East Asian Summer Monsoon rainfall reconstruction from the northwest Chinese loess plateau over the last 650 ka.

In this study the authors address the following questions: i) is there a reliable proxy for East Asian summer monsoon (EASM) rainfall at the millennial timescale and ii) what are the factors modulating the millennial monsoon variability (MMV)? Overall the manuscript is clearly structured, well written and both topic and objectives are suitable for Climate of the Past.

First of all as I am not a specialist in "wavelet analysis" I will leave the evaluation of this approach to reviewers more familiar with statistical methods. On the other hand, as a geologist working on Loess-Palaeosol Sequences (LPS) for a long time, I am surprised (not to say displeased) by the complete absence of data presenting the loess and palaeosol record on which is based the present study.

Re: Thanks for reminding us of adding the basic information for stratum lithology of Linxia loess section to evaluate our age model. We would like to revise Figure 2 in our manuscript and add relative information in result section. See the following figure.

[Figure]

Revised Figure 2 in manuscript, a) Strata and down-core variations of mean grain size, magnetic susceptibility, Ca/Ti and sedimentation rate against depth (brown in grain size based age model

and dark brown in loess-cave comparison based age model). Brown red, orange and yellow rectangles represent palaeosol layers, weakly pedogenic palaeosol in loess layers and loess layers, respectively. The timing of dash line and glacial-interglacial transition are control points of grain size based chronology; b) variations of mean grain size, Ca/Ti over last 650 ka and age model of Linxia loess section. Comparison of mean grain size and Ca/Ti in Linxia section with Sanbao-Hulu (Cheng et al., 2009, 2016) and benthic $\delta^{18}O$ stack (Lisiecki and Raymo, 2005). The dark brown squares, blue triangles and red dots represent the first (glacial-interglacial transition), second (precession cycles) and third (millennial-scale events) class age control points at the corresponding position of cave record, respectively (Sun et al., 2021a). Light blue bands donate the interglacial times.

This is a main concern and that alone would be for me a matter to reject this contribution. Indeed, this is the starting point of the study: before presenting and interpreting the variation in Ti/Ca ratio and grain size parameters along the 182 m of the Linxia record, the LPS itself should be exposed with a reasonable level of information. Even if the Linxia LPS was previously published in another paper by Guo et al in Catena (2021) this information should be provided in the present contribution because it is very important for the evaluation of the "age model" on which all the conclusions of the study are based.

Re: Yes, I total understand your concern about our new age model. Because we did not detailedly explain the errors and differences of our new age model in method section. We have already published the age model and stratum lithology of Linxia section in Sun et al.,'s paper (Sun et al., 2021, ESR). We did not make this very clearly in our paper. In Sun et al,,'s, five high-sedimentation-rate loess records including Linxia loess section on the Chinese Loess Plateau are investigated to assess East Asian monsoon variability at orbital and millennial timescales (Figure 1). The independent speleothem-based chronology for Chinese loess-paleosol sequences over the past 640 ka is established by correlating loess grain size (a winter monsoon proxy) to speleothem $\delta^{18}O$ (a summer monsoon proxy). We would like to revise Figure 2 in our manuscript and show these information.

[Figure]

Figure 1. Comparison of loess proxies (χ-red lines, mean-blue lines) with speleothem δ18O (purple). From top to bottom: (A) Mangshan (MS) (Wang et al., 2020); (B) Dingbian (DB); (C) Linxia (LX); (D) Gulang (GL); (E) Jingyuan (JY) (Sun et al., 2019; this study); and (F) Speleothem δ18O from Hulu and Dongge caves (Wang et al., 2008; Cheng et al., 2016). Light green bars denote interglacial stages (1-15). Dashed lines and purple diamonds (first-order time controls) denote correlations between loess/paleosol boundaries and glacial/interglacial transitions. Orange dots indicate the second-order time controls adopted for matching abrupt changes between loess mean and speleothem δ18O.

The second main concern is indeed the "age model". As Reviewer 1 I will ask: what is the implication of the age model errors for the wavelet coherence correlations that authors conducted (against GHG, ETP, Insolation and benthic d18O on Figure 4) and for the millennial-scale component extraction.

Re:We should take the influence of age model error into consideration in method section. We assessed the this kind of influence and add the result in method parts. We isolated the millennial-scale components of loess Ca/Ti at the different age model for comparison (Figure 2).

The age model difference of millennial-scale variability is 1-3 kyr. We applied Ca/Ti on grain size based age model to calculate the MMV and conduct wavelet coherence analysis, showing almost the same correlation and phase at the orbital frequency bands (Figure 3) as that of WTC results at loess-cave comparison based age model (manuscript Figure 4).

[Figure]

Figure 2 The comparison of millennial-scale loess Ca/Ti variations on grain size based age model (purple line) and loess-cave comparison based age model (orange line).

[Figure]

Figure 3 Comparison of a) ice volume, b) GHG forcing, c) ETP and d) local summer insolation with MMV of loess Ca/Ti over the past 650 ka at the grain size based age model.

Before to present and age model you should provide variations curves of climate proxies considered here regarding to depth (Ca/Ti and loess mean grain size). The age model proposed in the present contribution is directly extracted from the one published by Guo et al., 2021. Looking to Fig. 3 of this paper I can agree with the age-depth relation proposed for the last 60 ka where OSL dating are available but concerning the part of the record older than the Last Glacial no absolute data are provided and the age model is thus highly speculative.

Re: Yes, I agree with you. We know that $^{14}C$ and OSL dating method could provide us absolute dating results for loess record over past 130 ka. For older glacial periods extending back B/M boundaries, there is no good dating method to obtain absolute dates. In East Asian monsoon region, high-precision dated stalagmites from the Hulu and Sanbao caves offer high-resolution $\delta^{18}O$ records reflecting orbital-to-millennial Asian monsoon variability over the past 640 ka (Wang et al., 2001, 2008; Cheng et al., 2009, 2016). Since most of the U-Th dating errors are less than 2 ka for the last 450 ka and increase to 4-8 ka before 450 ka, the composite cave $\delta^{18}O$ record has been used as an important benchmark for regional-to-global synchronization of abrupt climate changes. During past two decades, optical simulated luminance (OSL) dating has been successfully applied for reconstructing an independent chronology of the last glacial loess deposits (e.g., Lu et al., 2007; Lai et al., 2007; Stevens et al., 2008). Abrupt changes in two OSL-dated loess grain-size time series match well with millennial-scale climate events recorded by stalagmites, marine and ice cores (NGIP), implying that all abrupt events during the last glaciation are coupled among these records (Sun et al., 2012; Wang et al., 2020). Proxy-model comparisons suggest that abrupt changes of the winter and summer monsoon events are anti-correlated at millennial and centennial timescales in response to the meltwater forcing and the resulting change in the Atlantic meridional overturning circulation (Sun et al., 2012; Wen et al., 2016). At glacial-interglacial time scale, numerous loess proxies have demonstrated that during glacial terminations remarkable strengthening of the summer monsoon was associated with synchronous weakening of the winter monsoon (e.g., Liu and Ding, 1998; Xiao et al., 1995; An, 2000). Therefore, an independent loess chronology can be generated by matching rapid changes in loess grain-size (a winter monsoon indicator) with abrupt shifts of the cave $\delta^{18}O$ record (a summer monsoon proxy) at times of glacial/interglacial transitions. We could make a comparison between the typical grain size based model and our new speleothem-based age model and evaluate the accuracy of this new speleothem-based age model. The high-precision dated stalagmites are proven to be successive. The comparison of the millennial-scale signals among loess, speleothem, marine records could also check whether erosion hiatus occur in the loess record. The abrupt climate events are well recorded by loess MGS. The magnitude of millennial-scale abrupt climate events in loess MGS are equal or more than that of cave $\delta^{18}O$.

[Figure]

Figure 4 Correlation of abrupt climate events in terrestrial, marine and ice-core records. (A) Si/Sr ratio and its high-frequency components of IODP site 1308 (Hodell et al., 2008); (B) LGS640 and its millennial component; (C) Cave $\delta^{18}O$ and its millennial variability (Wang et al., 2008; Cheng et al., 2016); and (D) A synthetic Greenland temperature (GLT) and its high-frequency component (Barker et al., 2011). Abrupt climate events are recognized based upon their amplitude (>average deviation) and duration (>0.8 ka) of high-frequency components of these records. Purple and pink numbers above red curves indicate the warm DO-like events during each glacial and interglacial.

Deep blue numbers below the IODP 1308 Si/Sr curve denote the Heinrich-like events. (Sun et al., 2021, ESR, manuscript Figure 8).

The age-model is classically build using "tie points" that can be selected by "matching the loess (L)/paleosol (S) boundaries to the glacial/interglacial transitions. This is the classical approach but they should know that it is only reliable if: 1) the sedimentation rates are more or less regular through time during each glacial period and 2) no erosion hiatus occur in the record. This is not the case for the Linxia LPS according to Fig. 3 published by Guo et al. 2021.

For example it is strange to note that the sharp boundary in MGS and MS data occurring at the base of L2SS2 soil (-60m) has no counterpart in MIS stratigraphy. Following the correlation methodology exposed above (tie points) this major limit would rather have been correlated to the base of MIS 7 interglacial and thus dated at about 220 ka and not at about 175 ka according to the present scheme. In addition why the Last interglacial (MIS 5e) is only marked by a short and relatively not intense peak in both MGS and Ca/Ti curves (SISS3) whereas it is generally represented by a thick and well developed soil horizon in LPS?.

Re: Yes, it is strange. I think it is because of proxy sensitivity difference to climate factors forcing. As to LR04 record (mostly ice volume forcing), it is a good target to provide the glacial-interglacial age control points. The precession scale variations are not obvious as that preserved in Cave $\delta^{18}O$ record (mainly strong summer insolation forcing), especially during the glacial times. Due to weak pedogenesis and high sedimentation rates, precession- to millennial-scale oscillations are well preserved in the western and northwestern Chinese Loess Plateau (CLP) over the past glacial cycles (Sun et al., 2012, 2021a; Ma et al., 2017; Guo et al., 2021, Figure 5). However, these signals are not clear in middle and southern CLP. For example, the precession-scale variations are evident in Linxia Ca/Ti, but not in that of Galang Ca/Ti during MIS3 and MIS 6, which is caused by precipitation difference across the CLP. During the MIS 5e, the prevalent stronger East Asian monsoon results in intensified precipitation and leaching effect associated with flat low Ca/Ti value (low Ca/Ti reflecting stronger precipitation-induced leching, such MIS 11c,d and MIS 13a). While weaker winter monsoon leads to deposition of smaller particle size corresponding to low value of MGS.

[Figure]

Figure 5 Comparison of records of Ca/Ti, Rb/Sr and Zr/Rb determined by the CXRF method, magnetic susceptibility (MS), and mean grain size (MGS) for the Linxia (LX) and Gulang (GL, Sun et al., 2016) sections with the Chinese Sanbao-Hulu speleothem δ18O (Wang et al., 2008; Cheng et al., 2009, 2016) and the normalized stack loess median diameter (SLMD) records (Yang and Ding, 2014). Light blue bars indicate precessional cycles of different proxies corresponding to to July insolation maxima (Berger, 1978). Black numbers are DO (warming) events.

In addition the period corresponding to MIS 3 (± 60-30ka) exhibits only two weakly developed soil horizons (L1 SS1 and L1 SS2). These soils are likely corresponding to composite (upbuilding) soils developed over quite long periods (± 40-26 ka for L1SS1 and 60-45 ka for L1SS2) and thus including numerous DO events. So, the response to millennial timescale climate variations is clearly not recorded in the stratigraphic signal. In addition many of the peaks in proxies are so thin

and of so small amplitude (MGS variation ≤ to 3 mm) that interpreting them as the result of a DO event is very difficult to support.

Re: Yes, these two sub-paleosoil are formed during MIS 3. If we just take the absolute value of millennial-scale MGS variations into consideration, the value is not large. The absolute value of glacial-interglacial MGS variations varies from 7 to 10 μm. 2-4 μm fluctuations of millennial-scale MGS relative to orbital-scale variations are pretty large. Abrupt climate change is commonly defined as a transition in the Earth's climate system whose duration is short relative to the duration of the preceding or subsequent state (Overpeck and Cole, 2006). The high-pass filtering (10 kyr) components of MGS presented in Figure 2 are well correlated with abrupt climate events in Cave $\delta^{18}O$ and synthetic Greenland temperature (GLT) record (Figure 4). The amplitude of some abrupt climate events are relative small in MGS record, but it is clear.

Finally the authors seem to ignore that millennial time-scale climate variations have been evidenced for more than 20 years in European loess series (e.g. Antoine et al., 2001 (QI), 2009 (QSR); Moine et al., 2008 (QI); Rousseau et al, 2002 (QSR), 2017 (QSR); 2020, (CP); 2021 (QSR) and that they have been definitely dated and correlated with NGIP record using [14]C dating (Moine et al., 2017(PNAS) and Ujvary et al., 2017 (PNAS).

Re: Thanks for minding us of millennial time-scale climate variations recorded in European loess sequences. They are good articles and worthy of reading. Our group will constantly pay attention to loess research in Europe. These researches confirmed that millennial-scale climate variations could be well preserved in high resolution eolian loess record of Europe and China. They are synchronous with abrupt climate events documented in high latitude ice core and marine records.

Conclusion: The authors must answer to the major comments exposed above to demonstrate that the main conclusions of their paper can be fully supported by data before publication.
Re: Thanks for giving useful suggestion to improve our manuscript and let reader knows more about this special geological archive. Any further advice and question are welcomed.

---

## Author Response (AR1)

Dear Editor and Referees,

Thanks for your valuable comments for our paper. We revised the manuscript accordingly.

Editor Comments:

Many thanks for your submission and for the answers to the comments of the reviewer. I note that the reviewers have serious question on the influence of the uncertainty on the chronology of your record for the conclusions that you discuss here. As a consequence, I encourage you to submit a revised version of the manuscript with a significant section dedicated to the evaluation of the chronology and of the associated uncertainties in the revised manuscript. I also encourage you to highlight the periods when the chronology is uncertain as noted by the second reviewer and to quantify the influence of the uncertainties on the chronology on the wavelet coherence correlations. Many thanks for your efforts.

Re: We have addressed the chronology and the associated influence of the chronology uncertainty on the wavelet coherence results in the method section. See lines 117-129. We assessed the WTC result on the basis of the two age models published in Sun et al., (2021a, ESR). While there are differences in the two age models (correlation to marine $\delta^{18}O$ versus to speleothem $\delta^{18}O$), the MMV and associated wavelet coherence (WTC) are insensitive to these differences. Because the age model tie points are separated by 20-30 kyr, the differences in the two age models do not influence the the amplitude of the millennial-scale variability in the records. Only minor differences in phase and correlation of MMV WTC results are observed at the obliquity and precession bands (compare Figure 1 below to manuscript Figure 4). This is why we did not emphasize the influence of age uncertainties in the initial paper. We have included Figure 1 below as supplementary Figure S1 and address the age uncertainties in the method section. The WTC results (correlation and phase) are insensitive to changes in the associated orbital-scale age control unless errors are increased to approximately 10 kyr or greater. The second reviewer also suggests we include the Linxia loess/paleosol stratigraphy; we have included it in Fig. 2 and captions. See lines 486-491.

First Referee Comments:

Guo et al., generated a high-resolution summer monsoon proxy (Ca/Ti) from Linxia on the western CLP. The Ca/Ti ratio is interpreted as a precipitation-sensitive proxy linked to summer monsoon rainfall (Guo et al., 2021). The new precipitation proxy (Ca/Ti) and an East Asian composite speleothem $\delta^{18}O$ record ($\delta^{18}Osp$) are evaluated to elucidate the modulating drivers of these two proxy records.

The authors find that the MMV of Ca/Ti is mainly modulated by ice volume and greenhouse gases (GHG) at the eccentricity band. Both GHG and summer insolation modulate the MMV of Ca/Ti at the precession band but not that of $\delta^{18}Osp$; $\delta^{18}Osp$ MMV is modulated by winter insolation at the eccentricity and obliquity bands. The inferred mechanism of how these internal and external factors modulate the MMV calls on dynamic linkages to variability in AMOC at both eccentricity and precession bands.

Results suggest that the MMV EASM rainfall is modulated by ice volume, GHG, and insolation factors, consistent with those predicted to influence future changes in monsoonal precipitation.

This is an interesting study in which the authors address two important outstanding questions: 1) is there a reliable proxy for East Asian summer monsoon (EASM) rainfall at the millennial timescale and 2) what factors modulate the MMV thereof?

The paper is clear and well wrote, and is suitable for a journal such as Climate of the Past. However the authors must answer to major/minor comments (see below) to be sure that the main conclusions of their paper can be fully supported before considering publication.

Re: Thanks for reviewing our paper and giving us useful advice to improve our manuscript.

Major comment:

- There is no Figure with the age control points that include the error bars on these control points and more generally no errors for the age model used in this study. I suggest to add a figure that include the depth/age and errors for the age model of the loess record.

Re: The detail information of the age model was published in Sun et al., (2021a, Earth-Science Reviews). The reference is now cited and discussed in detail in the methods section. We revised Fig.2 in our manuscript accordingly to show the age difference and uncertainties for two difference age model. See lines 117-121. We stress, however, that the analyses in this manuscript are insensitive to small age model differences.

What is the implication of the age model errors for the wavelet coherence correlations that authors conducted (against GHG, ETP, Insolation and benthic $\delta^{18}O$ on Figure 4) and for the millennial-scale component extraction (Figure 3) ?

Re: Comparison of Figure 1 below with manuscript figure 4 demonstrates that the age model

uncertainties are of little to no consequence to the results and interpretation of the WTC analyses.

[Figure]

Figure 1 Comparison of a) ice volume, b) GHG forcing, c) ETP and d) local summer insolation with MMV of loess Ca/Ti over the past 650 ka on benthic δ¹⁸O tuning age model for comparison to Figure 4 in manuscript .

Minor comments:

- What is the resolution of the Ca/Ti record (in years) before resampling? I could not find it in the text.

Re: Sorry for forgetting to add the resolution of our new proxy. We revised it accordingly. See line 98. The powder samples were scanned at 2 cm interval, with the resolution ranging 10~200 yr/cm (revised the Fig.2 in our manuscript).

- Introduction part (lines 51-60) : "flood and drought events". What is the definition of the authors here for flood and drought events? And at which time scale this events occur? Are they directly related to the millennial scale variability the authors reconstruct in this paper?

Re: In this paper, the definition of flood and drought events are centennial to decade (100-10 yr) timescale natural disaster.

Second Referee Comments:

The paper by Guo et al. submitted to CP is based on a new East Asian Summer Monsoon rainfall reconstruction from the northwest Chinese loess plateau over the last 650 ka.

In this study the authors address the following questions: i) is there a reliable proxy for East Asian summer monsoon (EASM) rainfall at the millennial timescale and ii) what are the factors modulating the millennial monsoon variability (MMV)? Overall the manuscript is clearly structured, well written and both topic and objectives are suitable for Climate of the Past.

First of all as I am not a specialist in "wavelet analysis" I will leave the evaluation of this approach to reviewers more familiar with statistical methods. On the other hand, as a geologist working on Loess-Palaeosol Sequences (LPS) for a long time, I am surprised (not to say displeased) by the complete absence of data presenting the loess and palaeosol record on which is based the present study.

Re: Thanks for reminding us of adding the basic information for the loess/paleosol lithology of Linxia loess section. We revised Figure 2 in our manuscript and add relevant information and modified figure caption accordingly.

This is a main concern and that alone would be for me a matter to reject this contribution. Indeed, this is the starting point of the study: before presenting and interpreting the variation in Ti/Ca ratio and grain size parameters along the 182 m of the Linxia record, the LPS itself should be exposed with a reasonable level of information. Even if the Linxia LPS was previously published in another paper by Guo et al in Catena (2021) this information should be provided in the present contribution because it is very important for the evaluation of the "age model" on which all the conclusions of the study are based.

Re: Yes, I completely total understand the concern regarding details of the age model. The detailed age model is published in Sun et al., (2021, ESR). We cited this paper in the method section (previous version, lines 105-106), but did not make this very clear. The Sun et al., (2021a) paper analyzes five high-sedimentation-rate loess records including the Linxia loess section on the Chinese Loess Plateau to assess East Asian monsoon variability at orbital and millennial timescales (See Fig. 2 in Sun et al., 2021a, ESR). The independent speleothem-based chronology for Chinese loess-paleosol sequences over the past 640 ka is established by correlating loess grain size to speleothem $\delta^{18}O$. We have revised Figure 2 in our manuscript to show the requested information. See lines 104-116.

The second main concern is indeed the "age model". As Reviewer 1 I will ask: what is the implication of the age model errors for the wavelet coherence correlations that authors conducted (against GHG, ETP, Insolation and benthic d18O on Figure 4) and for the millennial-scale component extraction.

Re: We now make it clear in the revised manuscript that the analyses conducted here are insensitive to differences in orbital scale age models – see response to Rev. 1 above.

Before to present and age model you should provide variations curves of climate proxies considered here regarding to depth (Ca/Ti and loess mean grain size). The age model proposed in the present contribution is directly extracted from the one published by Guo et al., 2021. Looking to Fig. 3 of this paper I can agree with the age-depth relation proposed for the last 60 ka where OSL dating are available but concerning the part of the record older than the Last Glacial no absolute data are provided and the age model is thus highly speculative.

Re: This age model is based on commonly accepted methodologies for applying chronologies to loess/paleosol profiles from the Chinese loess plateau (See, for example, Beck et al., 2018, Science; Sun et al., 2021a, ESR; Zhang et al., 2022, GRL). We briefly review the age model philosophy and evolution below but note that our analytical approach is insensitive to small age uncertainties in this sequence.

We know that $^{14}$C and OSL techniques provide absolute dating results for loess sections over the past 130 ka. For older glacial periods extending back B/M boundaries, there is no dating method to obtain absolute dates. In the East Asian monsoon region, high-precision (U-Th) dated stalagmites from the Hulu and Sanbao caves offer high-resolution $\delta^{18}$O records reflecting orbital-to-millennial Asian monsoon variability over the past 640 ka (Wang et al., 2001, 2008; Cheng et al., 2009, 2016). Hence, the composite cave $\delta^{18}$O record has been used as an important benchmark for regional-to-global synchronization of abrupt climate changes (Beck et al., 2018, Science; Sun et al., 2021a, ESR; Zhang et al., 2022, GRL). During the past two decades, optical simulated luminance (OSL) dating has been successfully applied for reconstructing an independent chronology of the last glacial loess deposits (e.g., Lu et al., 2007; Lai et al., 2007; Stevens et al., 2008), and is known to be consistent with the cave $\delta^{18}$O chronologies; Abrupt changes in two OSL-dated loess grain-size time series match well with millennial-scale climate events recorded by stalagmites, marine and ice cores (NGIP), implying that all abrupt events during the last glaciation are coupled among these records (Sun et al., 2012; Wang et al., 2020). Proxy-model comparisons suggest that abrupt changes of the winter and summer monsoon events are anti-correlated at millennial and centennial timescales in response to the meltwater forcing and the resulting change in the Atlantic meridional overturning circulation (Sun et al., 2012; Wen et al., 2016). At glacial-interglacial time scale, numerous loess proxies have demonstrated that during glacial terminations remarkable strengthening of the summer monsoon was associated with synchronous weakening of the winter monsoon (e.g., Liu and Ding, 1998; Xiao et al., 1995; An, 2000). Therefore, an independent loess chronology can be generated by matching rapid changes in loess grain-size (a winter monsoon indicator) with abrupt shifts of the cave $\delta^{18}$O record (a summer monsoon proxy) at times of glacial/interglacial transitions. This approach not only improves the accuracy of the loess/paleosol age model but also helps to demonstrate the continuity of the Chinese loess plateau sections as well, See Sun et al 2021a, Figure 2 below.

[Figure]

Figure 2 Correlation of abrupt climate events in terrestrial, marine and ice-core records. (A) Si/Sr ratio and its high-frequency components of IODP site 1308 (Hodell et al., 2008); (B) LGS640 and its millennial component; (C) Cave δ¹⁸O and its millennial variability (Wang et al., 2008; Cheng et al., 2016); and (D) A synthetic Greenland temperature (GLT) and its high-frequency component (Barker et al., 2011). Abrupt climate events are recognized based upon their amplitude (>average deviation) and duration (>0.8 ka) of high-frequency components of these records. Purple and pink numbers above red curves indicate the warm DO-like events during each glacial and interglacial.

Deep blue numbers below the IODP 1308 Si/Sr curve denote the Heinrich-like events. (Sun et al., 2021, ESR, manuscript Figure 8).

The age-model is classically build using "tie points" that can be selected by "matching the loess (L)/paleosol (S) boundaries to the glacial/interglacial transitions. This is the classical approach but they should know that it is only reliable if: 1) the sedimentation rates are more or less regular through time during each glacial period and 2) no erosion hiatus occur in the record. This is not the case for the Linxia LPS according to Fig. 3 published by Guo et al. 2021.

For example it is strange to note that the sharp boundary in MGS and MS data occurring at the base of L2SS2 soil (-60m) has no counterpart in MIS stratigraphy. Following the correlation methodology exposed above (tie points) this major limit would rather have been correlated to the base of MIS 7 interglacial and thus dated at about 220 ka and not at about 175 ka according to the present scheme. In addition why the Last interglacial (MIS 5e) is only marked by a short and relatively not intense peak in both MGS and Ca/Ti curves (SISS3) whereas it is generally represented by a thick and well developed soil horizon in LPS?.

Re: Yes, it is interesting. I think it is because of proxy sensitivity differences. The LR04 record (mostly ice volume forcing), is a good target to provide the glacial-interglacial age control points but the precession scale variations are not nearly as strongly recorded as in the Cave $\delta^{18}O$ record (mainly strong summer insolation forcing), especially during the glacial times. Due to weak pedogenesis and high sedimentation rates, precession- to millennial-scale oscillations are well preserved in the western and northwestern Chinese Loess Plateau (CLP) over the past glacial cycles (Sun et al., 2012, 2021a; Ma et al., 2017; Guo et al., 2021, Figure 3 below). However, these signals are not clear in middle and southern CLP. For example, the precession-scale variations are evident in Linxia Ca/Ti, but not in that of Galang Ca/Ti during MIS 3 and MIS 6, which is caused by precipitation difference across the CLP (Figure 3 below). During the MIS 5e, the prevalent stronger East Asian monsoon results in intensified precipitation and leaching effect associated with flat low Ca/Ti value (low Ca/Ti reflecting stronger precipitation-induced leaching, such MIS 11c,d and MIS 13a). While weaker winter monsoon leads to deposition of smaller particle size corresponding to low value of MGS.

[Figure]

Figure 3 Comparison of records of Ca/Ti, Rb/Sr and Zr/Rb determined by the CXRF method, magnetic susceptibility (MS), and mean grain size (MGS) for the Linxia (LX) and Gulang (GL, Sun et al., 2016) sections with the Chinese Sanbao-Hulu speleothem δ¹⁸O (Wang et al., 2008; Cheng et al., 2009, 2016) and the normalized stack loess median diameter (SLMD) records (Yang and Ding, 2014). Light blue bars indicate precessional cycles of different proxies corresponding to to July insolation maxima (Berger, 1978). Black numbers are DO (warming) events.

In addition the period corresponding to MIS 3 (± 60-30ka) exhibits only two weakly developed soil horizons (L1 SS1 and L1 SS2). These soils are likely corresponding to composite (upbuilding) soils developed over quite long periods (± 40-26 ka for L1SS1 and 60-45 ka for L1SS2) and thus including numerous DO events. So, the response to millennial timescale climate variations is clearly not recorded in the stratigraphic signal. In addition many of the peaks in proxies are so thin

and of so small amplitude (MGS variation ≤ to 3 mm) that interpreting them as the result of a DO event is very difficult to support.

Re: Yes, these two sub-paleosols are formed during MIS 3. If we just take the absolute value of millennial-scale MGS variations into consideration, the value is not large. The absolute value of glacial-interglacial MGS variations varies from 7 to 10 μm. 2-4 μm fluctuations of millennial-scale MGS relative to orbital-scale variations are pretty large. Abrupt climate change is commonly defined as a transition in the Earth's climate system whose duration is short relative to the duration of the preceding or subsequent state (Overpeck and Cole, 2006). The high-pass filtering (10 kyr) components of MGS presented in Figure 2 are well correlated with abrupt climate events in Cave $\delta^{18}$O and synthetic Greenland temperature (GLT) record (Figure 3). The amplitude of some abrupt climate events are relatively small in MGS record, but it is clear.

Finally the authors seem to ignore that millennial time-scale climate variations have been evidenced for more than 20 years in European loess series (e.g. Antoine et al., 2001 (QI), 2009 (QSR); Moine et al., 2008 (QI); Rousseau et al, 2002 (QSR), 2017 (QSR); 2020, (CP); 2021 (QSR) and that they have been definitely dated and correlated with NGIP record using $^{14}$C dating (Moine et al., 2017(PNAS) and Ujvary et al., 2017 (PNAS).

Re: Thanks for reminding us of millennial time-scale climate variations recorded in European loess sequences. They are good articles and worthy of reading. Much has been learned from loess research in Europe. In the same manner that $^{14}$C dating has been utilized to verify correlations of the European loess to NGRIP, $^{14}$C and OSL have been utilized to verify correlation of East Asian loess to speleothem $\delta^{18}$O. To better understand long-term climate change preserved in East Asian sequences, these correlations are applied beyond the limits of $^{14}$C and OSL dating to derive Pleistocene time-scales as described above.

Conclusion: The authors must answer to the major comments exposed above to demonstrate that the main conclusions of their paper can be fully supported by data before publication.
Re: Thanks for providing useful suggestion to improve our manuscript. We hope we have addressed reviewer concerns sufficiently.

Yours sincerely,

Fei Guo